# Marine Antimicrobial Peptides-Based Strategies for Tackling Bacterial Biofilm and Biofouling Challenges

**DOI:** 10.3390/molecules27217546

**Published:** 2022-11-03

**Authors:** Anupam Patra, Jhilik Das, Nupur Rani Agrawal, Gajraj Singh Kushwaha, Mrinmoy Ghosh, Young-Ok Son

**Affiliations:** 1Transcription Regulation Group, International Centre of Genetic Engineering and Biotechnology (ICGEB), Aruna Asaf Ali Marg, New Delhi 110067, India; 2Vision Lab Centre for Neuroscience, Indian Institute of Science, Bangalore 560012, India; 3KIIT-Technology Business Incubator, Kalinga Institute of Industrial Technology (KIIT-DU), Bhubaneswar 751024, India; 4Department of Animal Biotechnology, Faculty of Biotechnology, College of Applied Life Sciences, Jeju National University, Jeju-si 63243, Korea; 5Department of Biotechnology, School of Bio, Chemical and Processing Engineering (SBCE), Kalasalingam Academy of Research and Educational, Krishnankoil 626126, India; 6Interdisciplinary Graduate Program in Advanced Convergence Technology and Science, Jeju National University, Jeju-si 63243, Korea; 7Bio-Health Materials Core-Facility Center, Jeju National University, Jeju-si 63243, Korea

**Keywords:** biofouling, biofilm, antimicrobial peptides, AMPs, natural peptides, marine source, antibacterial

## Abstract

An assemblage nexus of microorganisms enclosed in a composite extracellular polymeric matrix is called as a biofilm. The main factor causing biological fouling, or biofouling, is biofilms. Biofilm-mediated biofouling is a significant detrimental issue in several industries, including the maritime environment, industrial facilities, water treatment facilities, and medical implants. Conventional antibacterial remedies cannot wholly eradicate bacterial species owing to the structural rigidity of biofilm and the eventual growth of antibiotic-resistant microorganisms. Consequently, several approaches to disrupt the biofilm have been investigated to address this particular phenomenon. Antimicrobial peptides (AMPs) have emerged as a promising contender in this category, offering several advantages over traditional solutions, including broad-spectrum action and lack of antibiotic resistance. Because biofouling significantly impacts the marine industry, AMPs derived from marine sources may be suitable natural inhibitors of bacterial proliferation. In this article, we discuss the range of physicochemical and structural diversity and the model of action seen in marine AMPs. This makes them an appealing strategy to mitigate biofilm and biofilm-mediated biofouling. This review also systematically summarizes recent research on marine AMPs from vertebrates and invertebrates and their industrial significance, shedding light on developing even better anti-biofouling materials shortly.

## 1. Introduction

Biofilm, a naturally occurring matrix structure of immense microbial diversity, is formed by microorganisms enclosed in the extracellular composite. Bacterial biofilm proliferation is accomplished on the surface, assisted by extracellular polymeric secretion [1]. These bacterial polymeric secretions enable them to exist on the surface and protect them from unfavorable and harmful exposure [2]. Bacterial morphology and physical properties as colloidal particles, such as size, general negative charge, and a range of growth rates, facilitate their invasion into various habitats [3]. Therefore, the mechanical complexity of biofilm formation proliferates as physical properties assist in becoming resistant to numerous chemical and radiation-based solutions [4].

Subsequent biofilm growth on the surface of objects gained complexity further and yielded harmful impact. Hence, the destructive effects of microorganisms on artificial materials are called biofouling. Biofouling substantially obstructs various processes by creating interferences, such as mechanical blockage (where engine performance decreases due to poor heat transmission), microorganism-based corrosion, product contamination, plumbing defect, medical contamination involving implants or prosthetic devices, marine pollution, oil pipeline blockage, cleaning equipment damage for hospitals and laboratories, and public health risks. (Figure 1).

Several solutions, including mechanical and chemical treatments, have been employed to address the critical challenge of biofilm-mediated biofouling. However, these solutions could be more effective. Hence, newer technologies with alternate solutions are being explored to overcome this adherent biofilm and biofouling. Recently, antimicrobial peptides (AMPs) have been shown to be effective in alleviating complex biofilm on the surfaces of various objects. AMPs are naturally abundant short peptides that are part of many species’ immune systems and have been extracted and identified from multiple sources. Antimicrobial Peptide Database (APD3) lists 3425 antimicrobial peptides from six life kingdoms (385 isolated/predicted bacteriocins/peptide antibiotics from bacteria, five from archaea, eight from protists, 25 from fungus, 368 from plants, and 2489 from animals) [5]. The amino acid length of AMPs varies significantly, and with up to 60 residues long polypeptide chains are considered peptides. Generally, cationic peptides exhibit excellent antimicrobial activity. Therefore, majorly reported AMPs are cationic, with an average charge of 3.32. However, some anionic AMPs are also reported which are composed of multiple repetitions of acidic amino acids, such as aspartic or glutamic acid [6,7,8]. The cationic AMPs are composed of basic charge-containing amino acid residues, such as arginine and lysine. Depending on the structural architecture, they potentially form α-helix, β-sheet, linear stretch, circular, and even more complicated bridge structures [9,10]. Their functional mode of action also varies significantly among various AMPs, particularly intervening intracellular processes. They can exhibit membrane damage and cellular injury [11], disrupt protein and DNA synthesis, and crosslink single or double-stranded DNA [12]. Several instances can sight upon the undeniable potency of AMPs that can block targeted cellular activity is yet to be discovered.

Besides the negative impact, biofilm-based biofouling has had the most significant impact on marine ecology. Because unicellular and multicellular animals thrive in these environments under extreme conditions, such as salt, pressure, and temperature, these ecosystems create the widest variety of AMPs in terms of novel sequences, structures, and antibiotic potential with a diverse spectrum of biomolecules. Hence, there is immense potential to explore the chemical and structural properties of such natural marine AMPs that may ease future concerns in the marine industry and human health [13]. When all the facts and details are considered, marine AMPs are a gold mine for biofilm-based biofouling control, with the ability to offer solution keys.

## 2. Biofilm Formation

The attachment of microorganisms initiates biofilm formation to the surfaces by employing various physicochemical interaction forces. Early colonies control colony expansion. Thus, the bacterial layer’s capacity to change surface characteristics is significant. A complex network with different species distribution, shapes, cell surface charges, and other unique cell-to-cell interaction patterns cannot predict community formation [14]. They may reduce strong hydrophobic interactions and increase hydrophilic contacts, affecting a surface’s adhesiveness [15]. The previous investigation shows how varied the substratum’s hydrophilicity, electrostatic interaction, and acid-base interaction are. Eleven of the sixteen bacteria had hydrophilic cell surfaces, while five had hydrophobic ones. They are negatively charged in typical saltwater because their isoelectric point is between 2.2 and 3.4, allowing electrostatic and acid-base interactions in specific strains [8]. Extracellular appendages such as pili and structural elements of attachment surfaces may help with initial attachments. There are four stages in forming a biofilm: (1) Biochemical conditioning: within minutes of submersion, macromolecules such as polysaccharides and peptides begin to adhere to the surface. (2) Bacterial colonization: bacteria begin to aggregate and form a layer after a few hours. (3) Unicellular eukaryotic colonization: secondary colonizers, such as diatoms, yeast, protozoa, and others, begin to settle on the colony after a few days, and eventually begins the process of (4) Multicellular eukaryotic colonization: multicellular eukaryotes such as larvae, spored, etc., entering the colony after a few weeks to a year [16,17].

Bacteria and diatoms dominate the population of marine biofilms. The proportion of bacteria, diatoms, and heterotrophic flagellates in the White Sea was 640:4:1 [17], while that of other single-celled creatures was just about 0.15%. [17,18]. According to a study, Roseobacter clade members are the most prevalent and dominating primary colonizers because of their quick response to nutritional conditions that cause film formation [19]. Since they are the original colonizers, they substantially influence the mature biofilm’s shape and function [20].

Bacterial cells have been shown to represent multicellular-like cluster formations in biofilms that have progressed via course of evolution. Those cluster representations may improve the survivability of predators and the supply of nutrients. Furthermore, biofilms can produce violacein, an antipredator secretion [21], and build a multilayered biofilm surface [14,21]. Biofilm develops into a self-sustaining complex colony that protects each participating cell from external stress such as temperature changes, pH shifts, dryness, radiation, food scarcity, predation, and other biological pressures.

## 3. Biofilm-Mediated Biofouling

The production of biofilms typically starts with biofouling, and the mechanical complexity of biofilms allows for biofouling maintenance. Most microfouling is initially produced by bacteria and diatoms, making space for other organisms like algae, fungi, barnacles, and others, known as macrofouling [16,22]. Depending on the surface layer, three types of biofilms develop at the start of fouling: type I has only bacterial cells in both living and dead states; type II has a diatom layer over the bacterial layer; and type III has a layer of multicellular organisms over the diatom layer [23] (Figure 2).

Biofilm buildup in ship hulls and pipelines especially submerged infrastructure has the most significant economic loss due to biofouling (Figure 1). Various approaches are currently being used to mitigate the financial loss mediated by biofouling. These include hull cleaning, sealing, and repainting. However, these approaches are not economical and cost more than $250 million annually [24]. Industrial equipment and engines with macrofouling significantly impact marine environments, potentially reducing heat transfer efficiency. It reduces water flow and increases biomass beneath aquaculture plants, increasing the risk of fish illness [25,26]. Since a coastal power plant requires a large amount of cooling water via submerged pipelines, biofouling impacts the smooth flow of water to heat exchangers. Subsequently, invertebrate larvae and other higher-order creatures breed on the heat exchanger walls, where a layer of bacteria has formed [18]. Macrofouling is influenced by local hydrodynamics and the environment. As a result of biofouling, the heat transfer module loses efficiency and becomes an insulating chamber. Biofouling also interferes with membrane separation systems used in water purification. Membrane biofilm increases membrane resistance during reverse osmosis, decreasing water flow.

## 4. Antimicrobial Peptides from Marine Sources

All living things contain natural antimicrobial peptides, which act as the first line of defense against microorganisms as part of their innate immune response [26,27]. Oceans are considered a repository for bioactive compounds as they make up more than 70% of our planet’s surface and more than half of its biodiversity [28]. Above all else, it offers versatile conditions for biodiversity, including fluctuations in pressure, salinity, illumination, and temperature that allow the diversification of AMPs molecules with various functional modes. AMPs are considered up to 60 amino acid-long peptide molecules. They typically have a net charge of +2 to +9 owing to the inclusion of lysine and arginine amino acids and hydrophobic residues, which facilitate successful adhesion to target pathogen membranes [29]. Counting on marine AMPs, the majority of them are cationic peptides, such as piscidin from teleost fish, aurelin from mesoglea of a scyphoid jellyfish, and Epinecidin-1, from fish [30,31] (Figure 3). At the same time, a small number of them are anionic and may fold into an amphipathic shape when they come into contact with membranes [32]. They display a wide variety of structural features, including α-helices, β-strands connected by disulphide bridges, loop, and extended structures, such as the α-helix peptide found in *Hediste diversicolor* and the antiparallel β-sheets protruding from CATH-BRALE, a salmonoid cathelicidin found in an ancient fish called *Bracgymysttazlenok* [33]. A wide variety of bacteria may be affected by their enormous structural diversity and other characteristics, such as size, charge, hydrophobicity, amphipathic stereo geometry, and peptide self-association to the biological membrane [34].

## 5. Mechanisms of Action

AMPs are contacted to target molecule surfaces primarily through electrostatic interaction. They penetrate cell membranes and obstruct vital cellular functions, including protein and nucleic acid synthesis, enzymatic reactions, and cell wall formation. AMPs are classified into two types based on their functional mode of action: membrane-acting and non-membrane-acting peptides.

### 5.1. Membrane Targeting Mechanism

The membrane targeting mechanism is primarily based on the membrane composition, particularly the lipid variation. Regarding biofilm production, glycerophospholipids, lysolipids, sphingolipids, and sterols are the major cell membrane lipids. Phosphatidylethanolamine (PE), phosphatidylglycerol (PG), and cardiolipin (CL) are prevalent lipids in bacteria, whereas phosphatidylcholine (PC), phosphatidylinositol (PI), phosphatidylethanolamine (PE), and phosphatidic acids (PA) are the primary glycerophospholipids in fungal cell membranes. Membrane-targeting antimicrobial peptides can break down biofilms by disrupting cell membranes. However, they differ from cell-penetrating peptides (CPPs), which can enter cells and enter through their membrane. CPPs fall short of the physicochemical standards for anti-biofilm peptides. Anti-biofilm peptides disrupt biofilm signals, permeate the cytoplasmic membrane and extra polymeric substance (EPS), alter EPS synthesis, and other mechanisms to attack biofilms, which can be used to treat bacterial infections that are persistent and multi-resistant. Another variation in the membrane targeting mechanism was reported as external membrane disruption [35]. Based on the current mode of action reported in the literature, the methods of action of AMPs can be further classified as follows (Figure 4).

#### 5.1.1. Toroidal Pore Model

The toroidal pore model, also known as the wormhole model, proposed by Matsuzaki and his colleagues. It describes how AMPs get trapped in the cell surface layer and bend to form a ring aperture with a radius of 1–2 nm [36]. These models’ common examples are arenicin, lacticin Q, and magainin 2. Additionally, cationic peptides like BP2, TC19, and TC84 weaken the membrane wall by forming fluid domains [37].

#### 5.1.2. Barrel-Stave Model

Multimers of antimicrobial peptides can penetrate a cell’s lipid bilayer and establish channels that lead to cytoplasmic outflow. AMPs may trigger apoptosis (cell death) in extreme situations [38], similarly to alamethicin, which uses mechanical structure to carry out its pore-forming function. Additionally, computations have demonstrated that both the inner and outer membranes of hairpin AMP protegrin-1 can generate stable octameric barrels and tetrameric arches (half barrels) [39].

#### 5.1.3. Carpet-like Model

Antimicrobial peptides are positioned along the cell membrane’s surface, with their hydrophobic ends directed toward the phospholipid bilayer and their hydrophilic ends oriented toward the solvent. AMPs will cover the membrane surface like a carpet and dissolve the cell membrane, acting as though they were employing detergent [40]. Significant amounts of AMPs are required, and a certain concentration threshold is required for this pore-forming process. This mechanism explains how the human cathelicidin LL-37 demonstrates its action, and how AMPs with a beta-sheet structure also play a part in this scenario [41]. When AMP cecropin P1 was applied flat to the surface of the pathogen’s cell membrane, it destabilized and ultimately destroyed the cell membrane, according to research using Fourier transform infrared spectroscopy (FTIR) [12].

### 5.2. Non-Membrane Targeting Mechanism

Once AMPs enter the cell, they activate a non-membrane target mechanism that suppresses cell division, nucleic acid synthesis, protein synthesis, and protease activity. In essence, AMPs have the potential to target any major cellular function. Bac7 1–35, Tur1A, and DM3, for example, may serve as protein synthesis inhibitors, with ribosomes or other pathways as possible targets [42]. AMPs also target inhibition of bacterial cell wall biosynthesis by lipid II binding, characteristic of some bacteriocins and defensins such as mersacidin, plantaricin, and nisin. Both pyrhocoricin and drosocin may disrupt protein maturation activity by inhibiting ATPase activity DnaK chaperone. Indolicidin, a C-terminally amidated cationic Trp-rich AMP with 13 amino acids and a particular target for the basic region of DNA, may block DNA topoisomerase I and crosslink single- or double-stranded DNA. It is an example of a nucleic acid biosynthesis inhibitor [43]. An AMP from tongues called TFP (Tissue factor pathway inhibitor) 1-1TC24 penetrates the cytoplasm of target cells upon cell membrane rupture, where it destroys DNA and RNA. Protease activity can prevent by histatin 5, eNAP-2, and indolicidin which can inhibit microbial serine proteases, elastase, and chymotrypsin; meanwhile, Cathelicidin-BF, a peptide isolated from the venom of *Bungarus fasciatus*, can successfully prevent thrombin-induced platelet aggregation and further block protease activated receptor 4 [44]. Due to their high DNA binding affinity and significant penetration ability, APP and MciZ may also stop cell division.

## 6. Classification of Marine AMPs

Oceans are home to various organisms, including mollusks, echinoderms, plants, algae, porifera, cnidaria, bacteria, and plants, among others, that are rich in biodiversity and provide a variety of sources of AMPs. The AMPs can be classified according to their (1) source, (2) charge, (3) structure or residual pattern, and (4) function. Here, we explain the source-oriented classification that helps to illustrate the vast diversity of the natural AMPs data set (Table 1). These AMPs display antimicrobial capabilities against bacteria, fungi, etc.

### 6.1. AMPs from Bacterial Sources

Bacteria synthesize AMPs in two modes: through bacteriocin (ribosomal) or the ribosomal independent pathway. Oceans harbor different types of bacteria, such as actinomycetes, proteobacteria, cyanobacteria, etc. Andrimid, cyclic dipeptides, cyclo-peptides, holomycin, indigoidine, kahalalides, massetolides, moiramide, ngercheumicins, solonamides, thiomarinols, unnarmicins, etc. are obtained from proteobacteria [45]. AMPs synthesized by non-ribosomal pathways include althiomycin and myxothiazols, and the list continues.

### 6.2. AMPs from Marine Invertebrates

Marine invertebrates produce around 40 different families of AMPs. Marine invertebrates live close to high bacterial densities, so they are rich sources of AMPs. 40% of the biomass of organisms belonging to phylum Porifera is attributed to bacteria, making them active producers of AMPs. A few peptides isolated from sponges include Callyaerin A and B from *Callyspongia aerizusa*, Theonellamide F and Theonellamide G isolated from *Theonella swinhoei*. Few peptides from sponges display anti-HIV activity: Koshikamides F and H from *Theonella cupola* and *T. swinhoei*, Celebesides A-C from *Siliquariaspongia mirabilis*, Mirabamides A–D from *Siliquariaspongia mirabilis*, Mirabamides E–H from *Stelletta* sp., and Stellettapeptins A and B from *Stelletta* sp., whereas peptides from *Geodia barrette* possess antifouling activity (Table 2).

The phylum Cnidaria comprises 13,000 species and contains an impressive array of physiologically active peptides, including corals, jellyfish, and anemones [46]. Aurelin peptide isolated from *Aurelia aurita* showed antimicrobial activity against gram-positive bacteria by blocking the potassium channel [47] (Table 2). Crude extract of six cnidarians *Carijoa riisei*, *Muriceopsis sulphurea*, *Neospongodes atlantica*, *Palythoa caribeorum*, *Plexaurella grandiflora*, and *Phyllogorgia dilatata* inhibited growth of bacteria responsible for common hospital infection. Pd-AMP1 isolated from *Phyllogorgia dilatata* was found effective in controlling *S. aureus* [48] (Table 2).

Several AMPs have been isolated and characterized from various animals of the Mollusca phylum, including octopus, squid, oyster, and snails. AMPs myticusin-1, mytichitin-CB and myticusin-beta have been isolated from the mussel *Mytilus coruscus.* Myticusin-1 and Myticusin-beta are active against both gram-positive and gram-negative bacteria. Mytichitin-CB extracted from the hemolymph acts against gram-positive bacteria; *Bacillus subtilis*, *S. aureus*, *Sarcina luteus*, and *Bacillus megaterium*, and against the fungi *C. albicans* and *Monilia albicans*. Octominin isolated from Octopus minor inhibits the growth of *C. albicans* via formation of pores on cell walls and increases oxidative stress in the cell [49]. Peptides RpdefB isolated from *Ruditapes philippinarum* and VpMacin from *Venerupis philippinarum* shows bactericidal activity by increasing membrane permeability. Myticin C, isolated from *Mytilus galloprovincialis* is an antiviral peptide that interferes with viral replication (Table 2).

**Table 2 molecules-27-07546-t002:** Anti-microbial peptides isolated from marine organisms.

Source	Peptides	References
Porifera peptides	Callyaerin A and B (antimicrobial)	[48]
	Theonellamide F(antimicrobial)	[49]
	Theonellamide G(antimicrobial)	[49]
	Koshikamides F and H (antiviral)	[50]
	Celebesides A-C (antiviral)	[50]
	Mirabamides A–D (antiviral)	[51]
	Mirabamides E–H (antiviral)	[51]
	Stellettapeptins A and B (antiviral)	[52]
	Barettin and 8,9-dihydrobarettin (antifouling)	[53]
	Barrettides A and B (antifouling)	[54]
Cnidaria peptides	Aurelin (antimicrobial)	[45]
	Pd-AMP1 (antimicrobial)	[46]
Mollusca peptides	Myticusin-1 (antimicrobial)	[55]
	Mytichitin-CB (antimicrobial)	[56]
	Myticusin beta (antimicrobial)	[57]
	Octominin (antimicrobial)	[47]
	RpdefB (antimicrobial)	[58]
	VpMacin (antimicrobial)	[58]
	Myticin C (antiviral)	[59]
Annelida peptides	Arenicin-1, 2, and 3 (antimicrobial)	[60]
	Hedistin (antimicrobial)	[61]
	Nicomicin -1 (antimicrobial)	[62]
	Capitellacin (antimicrobial)	[62]
	Perinerin (antimicrobial)	[63]
Arthropoda peptides	rSs-arasin (antimicrobial)	[64]
	Sphistin (antimicrobial)	[65]
	Anti-lipopolysaccharides (ALFs)(ALFPm11) (antimicrobial)	[66]
	Crustin (antimicrobial)	[67]
	paralithocins 1–3 (antimicrobial)	[68]
Echinodermata peptides	PpCrAMP (antimicrobial)	[69]
	SdStrongylocin 1 and 2 (antimicrobial)	[70]
	EeCentrocin 1 and 2 (antimicrobial)	[71]
	EeStrongylocin 2 (antimicrobial)	[71]
Chordata peptides	Pc-pis (antimicrobial)	[72]
	CodCath (antimicrobial)	[73]
	RbLEAP-2 (antimicrobial)	[74]
	Styelin D (antimicrobial)	[75]

Several AMPs, highly potent against bacteria and fungus, have been characterized by the Annelida phylum. Arenicin-1, 2, and 3 peptides have been isolated from polychaeta *Arenicola marina* lugworm. A broad-spectrum, bromotryptophan containing peptide, hedistin, was isolated from *Nereis diversicolor*. Nicomycin-1, a highly cationic peptide, was identified in the arctic polychaeta *Nicomache minor.* Capitellacin was reported from *Capitella teleta*. Perinerin, has been extracted from an Asian marine clamworm, *Perinereis aibuhitensis* Grube (Table 2).

Arthropoda phylum is an excellent source of AMPs as a diverse range of AMPs have been characterized by this phylum. Ss-arasin extracted from hemocytes of *Scylla serrate* showed significant activity against *S. aureus*, *P. aeruginosa*, and *E. coli*. Sphistin from *Scylla paramamosain* permeabilizes bacterial membranes of aquatic pathogens; *Aeromonas hydrophila*, *Pseudomonas fluorescens*, and *Pseudomonas stutzeri.* Crustin and Paralithocins 1-3 were isolated from *Penaeus monodon,* and *Paralithodes camtschaticus* display significant antimicrobial activities (Table 2).

Bioactive AMPs from marine invertebrates, Echinodermata, have been shown effective against various microorganisms. EeCentrocin was isolated from the coelomic fluid of the sea urchin *Echinus esculentus* and acted against *Corynebacterium glutamicum* and *S. aureus, E. coli* and *P. aeruginosa*. SdStrongylocin 1 and 2 inhibit *E. coli, S. aureus, C. glutamicum*, and *Listonella anguillarum*.

Among chordates, fishes are good sources of AMPs as they rarely get infected. Pleurocidins, Misgurins, chrysophsins, piscidins, moronecidins, and hepcidins are all obtained from fishes [73]. Pc-pis, Styelin D, codCath, and RbLEAP-2 were isolated from chordates; *Pseudosciaena croceaolated*, *Styela clava*, *Gadus morhua*, and *Oplegnathus fasciatus*, respectively.

### 6.3. AMPs from Marine Algae

Algae are a rich source of various bioactive compounds which show antimicrobial, antiviral, anti-tumorigenic, and antihypertensive properties. They produce a cocktail of peptides, polyphenols, alkaloids, polysaccharides, and fatty acids of pharmaceutical and industrial importance [76,77,78,79].

## 7. Strategies for AMPs Extraction from Marine Sources

Collection and isolation of bioactive peptides from marine sources is challenging due to the influence of seasonal availability, inhabitants, geographical location, and the ecological habitat of organisms [78,79,80]. Identification of a compound for antimicrobial activity begins with sample collection followed by monitoring the specific activities such as anti-bacterial, anti-fungal, anti-viral, etc. Isolation of bioactive peptides starts with the group of source organisms, followed by tissue isolation, trituration, or cell lysis. Extraction, concentration, precipitation, centrifugation, and membrane filtration are performed after this to remove particulate matter. Different chromatographic techniques, including size exclusion chromatography (SEC), ion-exchange chromatography (IEC), and solid-phase extraction (SPE), are used for further purification to remove the presence of inorganic compounds and fatty acids [81,82].

Mytilus defensins, mytilins, and mytimycin from hemocytes were isolated and purified using acetic acid extraction, C18 solid-phase extraction, reversed-phase high-performance liquid chromatography (RP-HPLC), size-exclusion chromatography, and further two rounds of RP-HPLC [55,56,79]. Perinerin from the whole clamworm was extracted in 1 M HCl + 5% formic acid + 1% TFA, then using SPE (C18), affinity chromatography (heparin), and lastly, RP-HPLC to get AMP [62]. Crustin (carcinin) was isolated from Shore crab hemocytes by 20% acetic acid, dialysis, ion exchange chromatography (cation), RP-HPLC, and finally, size exclusion chromatography (SEC) [83]. Arenicins from *Lugworm coelomocytes* were extracted in 10% acetic acid, followed by Ultrafiltration, AU-PAGE, and RP-HPLC [60]. Other marine AMPs undergo a similar multi-step purification process, the specifics of which are determined by the AMPs themselves [82,83]. The process of isolating and purifying AMPs is known as “bioassay-guided purification” since the presence of bioactive peptides is evaluated at each stage (Figure 5).

Alternate approaches are being explored to influence organisms for AMP production, such as exposing the organism to the bacterial pathogen [84]. Recombinant production of AMPs is another strategy that includes the extraction of mRNA from tissues of the organism and building a cDNA library. Identification of the AMP encoding ORF gene sequence was facilitated by primer-specific sequencing [85]. The recombinant production of hepcidin (CiHep) from grass carp (*Ctenopharyngodon idella*) was reported very recently [84]. The hepcidin-encoding gene was overexpressed in *Escherichia coli* and purified by affinity chromatography [84].

Recently, the chemical synthesis of marine AMPs has been explored to overcome the insufficiency of source organisms in bulk amounts. Total chemical synthesis of peptides offers advantages over isolation from marine organisms as it does not affect the biodiversity of an ecosystem. The total synthesis of cyclic peptides from the marine sponge *Stylissa carteri* has been described [85]. The synthesis of bioactive cyclic proline-rich heptapeptide through two-step solid-phase/solution synthesis offers a promising strategy for the bulk synthesis of natural AMPs. Chemical synthesis of highly effective AMP, anti-lipopolysaccharide factors (ALFs), was described recently [66]. This highly amphipathic peptide belongs to group G ALF from the Giant Tiger Shrimp (*Penaeus monodon*) (ALFPm11) [66].

Advancements in various research areas, such as developments in sequencing technologies, allow access to large data sets at low cost in minimal time, which could help quickly identify AMPs from marine organisms. Cutting-edge “Omics” technologies, including transcriptomics, proteomics, and metabolomics have eased the identification of drugs from natural sources. Integrating omics techniques with improved equipment, such as liquid chromatography-tandem mass spectrometry, and data mining platforms, potentially leads the way for innovative and better identification of potential AMPs from marine sources [86,87].

## 8. Advantages of Marine AMPs

In contrast to AMPs derived from terrestrial sources, marine AMPs provide a wide variety of different and exclusive chemical properties that consider giving them distinct advantages. They are particularly stable in high salt concentrations and may operate at low temperatures (4 °C to 20 °C) [88,89,90]. Though biofilms are collections of microbial communities with intricate, distinctive social virtues pertaining to defenses, natural marine AMPs may assault many targets in a biofilm to thwart its development. The positive aspects of natural marine AMPs are explained more below.

### 8.1. Exhibit Early-Stage Killing Potential

Biofilm undergoes fast entity changes as it develops. Therefore, AMP must be able to intervene quickly to impede growth, and we can avoid the drug resistance phenomena. Rapid application of AMPs to a targeted early-stage aids in preventing them from altering their phenotype and joining a stable community.

### 8.2. Act in Varied Micro-Environment and Niches in Biofilm

A biofilm’s environmental circumstances vary greatly. It is possible to see gradients in oxygen, nutrients, pH, and waste products. While cells at deeper places experience anoxia, nutritional deficiency, and acidic conditions, cells in the film’s perimeter enjoy an abundance of nutrients and oxygen. Diverse kinds of bacteria live in biofilm due to environmental variation. Metabolic heterogeneity is also evident in a similar group of microbes. Due to the film’s prevalent environmental gradations, a wide range of cells, from quickly dividing to slowly dividing to non-dividing cells known as “persisters”, are present. Combining marine AMP should effectively target the whole population, including persisters that are critical contributors to antibiotic resistance [91].

### 8.3. Hinder Cell Propagation through Extracellular Matrix Interference

The microbial community is encapsulated within an extracellular matrix (ECM) that acts as a physical barrier and provides protection. ECM comprises a self-produced matrix of extracellular polymeric substances (EPS). It also has proteins, lipids, and extracellular DNA (eDNA). It increases the stubbornness of the microbial community against antibiotics by delaying or even preventing interaction with microbes. Broad-spectrum AMPs can bind the active EPS component to inhibit its bio integrity. Besides, AMPs have proven their potential for pore formation in bacterial cells, which can be the ideal treatment for drug resistance [87,91].

### 8.4. Stands in the Way of Bacterial Communication

Bacterial cells communicate via auto-inducers and follow the higher virulence. For example, acyl homoserine lactones act as communication molecules in gram-negative bacteria. Still, in gram-positive bacteria, it is accomplished by quorum sensing (QS), which regulates the formation of biofilm and virulence traits of microorganisms. AMPs have the potency to serve as an antagonist of QS, which can solve the dual purpose of inhibiting biofilm formation and interfering with the virulence traits of the bacteria [92].

### 8.5. Synergistic Action with Other Antimicrobial Drugs

AMP must act synergistically with other antimicrobial drugs to combat bio-film, as none of the current medications can fight numerous traits associated with biofilm. Researchers across nations are trying to utilize the synergistic action of two or more antimicrobial agents to fight biofilm. Thus, an ideal AMP can extant its full strength in synergy with other antimicrobial drugs [92].

The successful applications of AMPs in various sectors, such as medicine, food, and veterinary, show great potential in other application sectors for AMPs [91]. Polylysine and nisin are already a preservative to inhibit the growth of microorganisms in food items. Since research on AMPs has been focused on therapeutical applications, several AMPs are already in the market. Bacitracin, colistin, fuzeon, and gramicidin are a few examples of AMPs that are being used clinically.

## 9. Conclusions

Current issues provide a window of opportunity when a tailored antimicrobial peptide may work successfully while still being environmentally benign. The process of producing biofilms is a complex environment with a wide variety of physiological factors and various stages for creating both micro and macro biofouling. In this situation, APMs may be an ideal approach for combating biofouling at an early stage when a biofilm forms; however, the delivery system beneath an open body of water should be more challenging. Modern technology, on the other hand, can isolate certain AMPs, and as for distribution, tagged coating, slow-dissolving tablets, spray, etc., may be done with correct engineering. Restoration of ecological balance is a sustainable strategy, and nature has many resources that can adequately handle the biofouling situation. Future maritime businesses should have considered this. According to earlier estimates, the maintenance costs associated with this specific issue should effectively decrease.

## Figures and Tables

**Figure 1 molecules-27-07546-f001:**
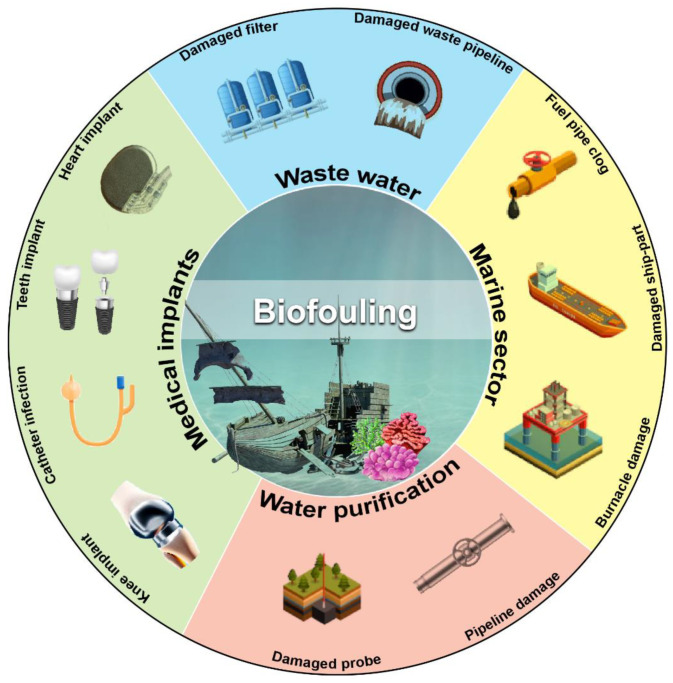
Schematic overview of the harmful impact of biofilm-mediated biofouling. Various sectors affected by biofouling are shown with representative pictures of the damaged part of affected objects. (Figure credit Freepik.com, accessed on 27 October 2022).

**Figure 2 molecules-27-07546-f002:**
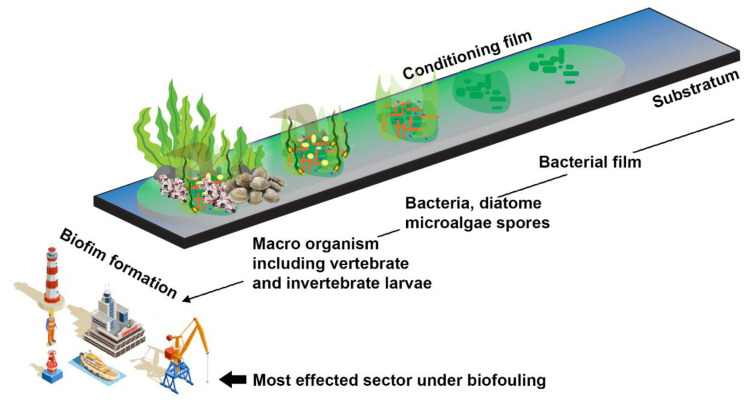
Major steps in biofilm formation may lead to biofouling events in the maritime sector. The biofouling started with the attachment of bacterial population on the object’s surface and was followed by the growth of other microorganisms, including diatoms, algae, and invertebrates. (Figure credit Freepik.com, accessed on 27 October 2022).

**Figure 3 molecules-27-07546-f003:**
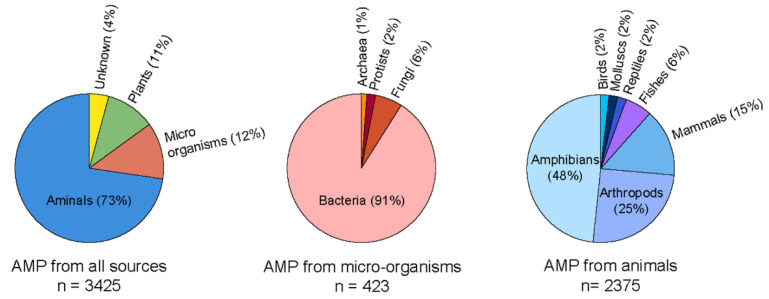
Current status of the natural AMPs to date as per AMP database. The circular pi charts were generated by data obtained from APD3 database [5].

**Figure 4 molecules-27-07546-f004:**
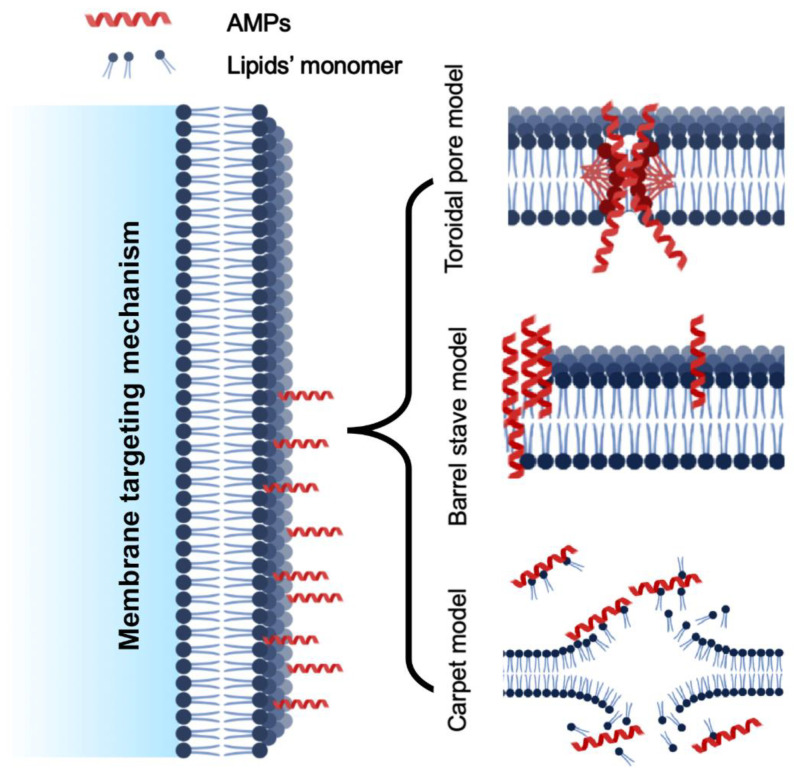
Various models of cell membrane targeting mechanism of AMPs.

**Figure 5 molecules-27-07546-f005:**
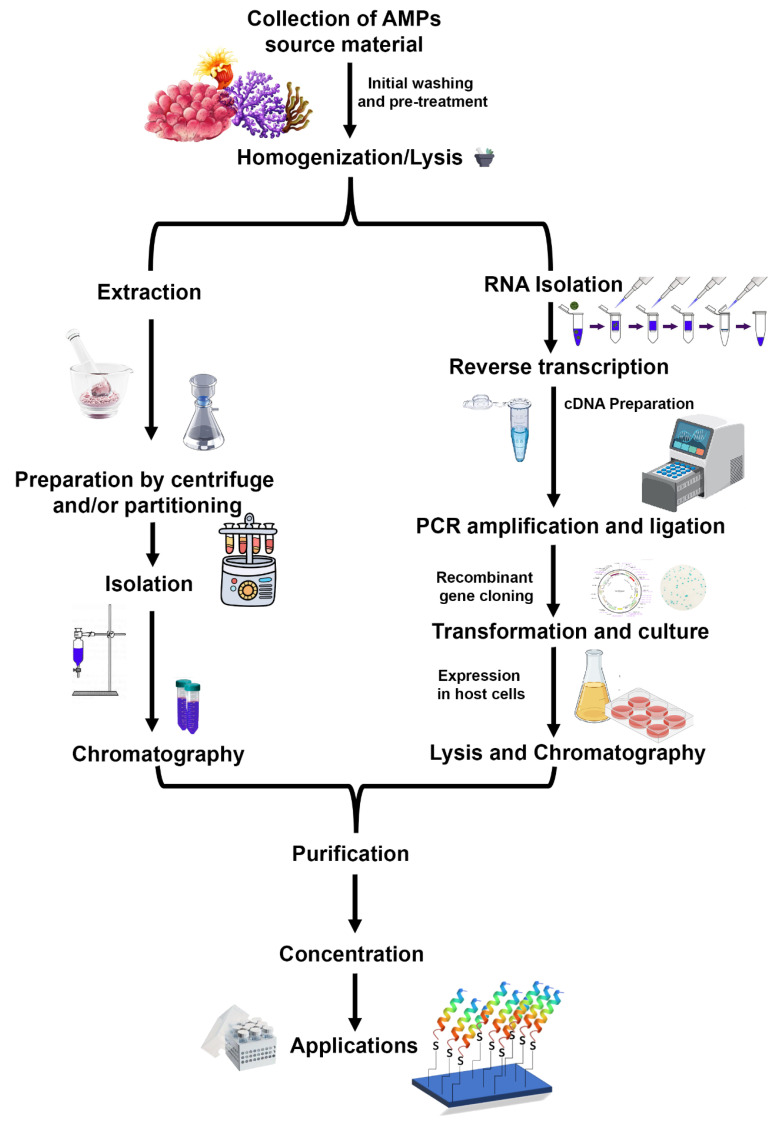
A generalized schematic workflow for the isolation and purification of antimicrobial peptides. A natural extraction from a native source employs extraction, partitioning, and chromatography steps, while recombinant peptide production includes mRNA isolation, cDNA preparation, cloning, and overexpression of peptide encoding gene. The recombinant peptide can be purified easily using affinity chromatography. (Figure credit Freepik.com, accessed on 27 October 2022).

**Table 1 molecules-27-07546-t001:** Summary of various types of AMPs and organized from data obtained from APD3 (As on 30 June 2022) [5].

Functional Type	Number of AMPs	Examples
Antibiofilm peptide	76	Pleurocidin, Nisin A, Gramicidin S
Antibacterial peptide	2900	Ericin S, Bactericidin B-3, Pleurocidin
Antifungal peptide	1257	Catfish PACAP38, Trematocine, Moronecidin-like
Antiparasitic peptide	140	Bombinin H4, HbbetaP-1, Piscidin 2
Insecticidal peptide	41	Magainin 2, Esculentin-1
Ion-channel inhibitor	7	Microcin H47, Bldesin
Protease inhibitor	33	Odorranain-B1, Microcin H47, Kunitzin-OS
Surface immobilized peptides	31	Magainin 2, Nisin A, Chrysophsin-1

## Data Availability

Not applicable.

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
