# Peer review of "Marine Antimicrobial Peptides-Based Strategies for Tackling Bacterial Biofilm and Biofouling Challenges"

_molecules, 2022, doi:10.3390/molecules27217546_

Round 1

Reviewer 1 Report

The manuscript is interesting, well written, well structured, the relevance of the topic is justified. There are no serious substantive claims, the comments and suggestions are mostly formal.

1. The reference numbers in the text do not correspond at all with the article numbers in the bibliography. One article in the reference list is duplicated (47&56). Please check carefully and correct the reference numbering.

2. The same as above applies to the numbering of figures.

3. There are many abbreviations in the text (for example, the names of phospholipids) which need to be specified. Sometimes there is a decryption, but it occurs in wrong place. E.g., EPS is mentioned first in Line 187 but specified only in Line 380. SPE is used along with "solid-phase extraction", but not specified as the same. Please check carefully and specify all abbreviations except for peptide names.

4. Give Latin names of species in italics throughout the text.

Minor points:

1. Eliminate the discrepancy in the size of the AMPs: 10-60 a.a. (Line 71) or up to 100 a.a. (Line 158)

2. Line 111: capitalize White Sea

3. Line 119: B biofilms - ???

4. Lines 181-182: PE and PC are zwitterionic lipids, not anionic ones.

5. Line 197: what do you mean by "later"?

6. I think it is reasonable to add to section 5.2. inhibition of bacterial cell wall biosynthesis by lipid II binding, which is characteristic of some bacteriocins and defensins.

7. Line 225: should be “drosocin”

8. Lines 246-247: ribosomal intermediate – maybe ribosomal independent?

9. Lines 328-330: reversed-phase high-performance liquid chromatography and RP-HPLC is the same.

Author Response

Reviewer comments and Response

 Reviewer #1

The manuscript is interesting, well-written, well structured, the relevance of the topic is justified. There are no serious substantive claims, the comments and suggestions are mostly formal.

Thank you for bringing this to our attention and also thankful to the reviewer for supporting us in further improving this manuscript. We have gone through the manuscript and carefully verify the comments received from the reviewer.

 Comment-1: The reference numbers in the text do not correspond at all with the article numbers in the bibliography. One article in the reference list is duplicated (47&56). Please check carefully and correct the reference numbering.

[Response] The text references, as well as the references in the tables, are completely modified and some more new references are also added to the content. The content in the manuscript has been modified carefully and sincerely.

Comment-2: The same as above applies to the numbering of figures.

[Response] We addressed the mistake and incorporated it into the document. We revise and rewrite the script.

Comment-3: There are many abbreviations in the text (for example, the names of phospholipids) that need to be specified. Sometimes there is a decryption, but it occurs in wrong place. E.g., EPS is mentioned first in Line 187 but specified only in Line 380. SPE is used along with "solid-phase extraction", but not specified as the same. Please check carefully and specify all abbreviations except for peptide names.

 [Response] We addressed the reviewers' concerns. The abbreviations in the text are specified accordingly. E.g., EPS is specified in line number 226, and SPE is specified in line number 410 along with the other specification.

Comment-4: Give Latin names of species in italics throughout the text.

[Response] The species name in italics throughout the text has been modified.

Minor points:

Comment-1: Eliminate the discrepancy in the size of the AMPs: 10-60 a.a. (Line 71) or up to 100 a.a. (Line 158).

 [Response] We addressed the reviewers' concerns.

 Comment-2: Line 111: capitalize White Sea.

[Response] We addressed the mistake and incorporated it into the document, Line number 141.

Comment-3: Line 119: B biofilms - ???.

[Response] We addressed the reviewers' concerns, and corrected this mistake (line 149).

Comment-4: Lines 181-182: PE and PC are zwitterionic lipids, not anionic ones.

[Response] The correction has been included (lines 217-223).

Comment-5: Line 197: what do you mean by "later"?

[Response] The correction has been included (line 268).

Comment-6: I think it is reasonable to add to section 5.2. inhibition of bacterial cell wall biosynthesis by lipid II binding, which is characteristic of some bacteriocins and defensins.

[Response] We modified the section and rewritten it (lines 294-300).

 Comment-7:  Line 225: should be “drosocin”

[Response] The correction has been included (line 300).

 Comment-8: Lines 246-247: ribosomal intermediate – maybe ribosomal independent?

[Response] We modified the section (line 325).

 Comment-9: Lines 328-330: reversed-phase high-performance liquid chromatography and RP-HPLC is the same.

[Response] We rectified the mistake and integrated it into the document (line 414).

Reviewer 2 Report

This manuscript presents a wealth of information related to antimicrobial peptides and applications, and describes the theory and process of biofilm formation, biofilm-mediated biofouling, marine-derived antimicrobial peptides, functional modes of action of AMPs, classification of marine AMPs, AMPs extraction strategies for marine resources, and the advantages of marine AMPs.

I suggest it would be great if the following information could be provided.

Since the title is Marine antimicrobial peptides-based strategies for tackling bacterial biofilm and biofouling challenges, chemical synthesis currently applies peptide synthesis, which should enable mass production of specific marine antimicrobial peptides, and it would be better if chemical synthesis and linking of them could be introduced.

Please indicate if there are any current examples of antimicrobial peptides being used in society.

It would be good to describe whether there are already industrial production examples of AMP, and apply it to relevant examples, including the price and scope of specific products. From this, we can understand the current development stage of AMP in industrial development and real-life applications.

The following message need to be clarified

Line 119 B biofilms can produce violacein, an antipredator secretion [22], B is not clear, be defined.

Figure 1 Knee implant should be included into the color of medical implant,

Line 182 PE, PG, and CL are prevalent anionic lipids in bacteria, whereas PC, PI, PE, and PA are the primary GPLs in fungal cell membranes.  Please define the PE, PG, CL, PC, PI, PE, PA and GPL.

Line 186 Anti-biofilm peptides disrupt biofilm signals, permeate the cytoplasmic membrane and EPS, alter EPS synthesis. Please define the EPS.

Line 243 These AMPs display antimicrobial capabilities against bacteria, fungi, etc. (Figure 3)  Figure 3 should be moved to Line 190

Line 190 Based on the current mode of action reported in the literature, the methods of action of AMPs can be further classified as follows (Figure 2). Figure 2 should be moved to biofilm formation section.

Line 270 Aurelin peptide isolated from Aurelia aurita showed antimicrobial activity against Gram-positive bacteria by blocking the potassium channel [49] (Table 1).  Table 1 should be changed to Table 2.

Line 286 Myticin C, isolated from Mytilus galloprovincialis is an antiviral peptide that interferes with viral replication (Table 1). Myticin C and above compounds cannot find in Table 1. Please check.

The Figure 4 should be explained and listed into appropriate section.

Figure 4. Current status of the natural AMPs to date as per AMP database (https://aps.unmc.edu/).

Line 377 8.3. Hinders production of extracellular matrix  You may consider hindering cell propagation through extracellular matrix interference 

Author Response

Reviewer #2

This manuscript presents a wealth of information related to antimicrobial peptides and applications, and describes the theory and process of biofilm formation, biofilm-mediated biofouling, marine-derived antimicrobial peptides, functional modes of action of AMPs, classification of marine AMPs, AMPs extraction strategies for marine resources, and the advantages of marine AMPs.

I suggest it would be great if the following information could be provided.

We appreciate the reviewer for supporting us in the further improvement of this manuscript. We carefully considered the comments and tried our best to address every one of them.

Comment-1: Since the title is Marine antimicrobial peptides-based strategies for tackling bacterial biofilm and biofouling challenges, chemical synthesis currently applies peptide synthesis, which should enable mass production of specific marine antimicrobial peptides, and it would be better if chemical synthesis and linking of them could be introduced.

[Response] We revise and rewrite the sentences throughout the manuscript to improve readability, and comprehension and make it easier to understand. We also added more explanations to rephrase various sections in the study and also added the additional figure for better presentation. 

Comment-2: Please indicate if there are any current examples of antimicrobial peptides being used in society.

 [Response] We are thankful to the reviewer for supporting us in the further improvement of this manuscript. To improve readability and comprehension, we revise and rewrite the entire sections of this manuscript.

Comment-3: It would be good to describe whether there are already industrial production examples of AMP, and apply it to relevant examples, including the price and scope of specific products. From this, we can understand the current development stage of AMP in industrial development and real-life applications.

 [Response] Nisin and polylysine AMPs are used as food preservatives to inhibit the growth of microorganisms. These are produced at an industrial scale in bulk quantities. Additionally, they are cost-effective and economical. Peptides are usually smaller compared to proteins, therefore, can be synthesized chemically. Thus, the chemical synthesis of peptides reduces production costs substantially.  

The following message need to be clarified

Comment-4: Line 119 B biofilms can produce violacein, an antipredator secretion [22], B is not clear, be defined.

 [Response] We addressed the reviewers' concerns, and corrected this mistake (line 149).

Comment-5: Figure 1 Knee implant should be included into the color of medical implant,

 [Response] Figure 1 has been modified accordingly.

Comment-6: Line 182 PE, PG, and CL are prevalent anionic lipids in bacteria, whereas PC, PI, PE, and PA are the primary GPLs in fungal cell membranes.  Please define the PE, PG, CL, PC, PI, PE, PA and GPL.

 [Response] We rectified the mistake and integrated it into the document. We have corrected the sentence (lines 217-224).

Comment-7: Line 186 Anti-biofilm peptides disrupt biofilm signals, permeate the cytoplasmic membrane and EPS, alter EPS synthesis. Please define the EPS.

 [Response] The correction has been included in the manuscript (line 226).

Comment-8: Line 243 These AMPs display antimicrobial capabilities against bacteria, fungi, etc. (Figure 3)  Figure 3 should be moved to Line 190

 [Response] We rectified the mistake and it has been corrected.

Comment-9: Line 190 Based on the current mode of action reported in the literature, the methods of action of AMPs can be further classified as follows (Figure 2). Figure 2 should be moved to biofilm formation section.

 [Response] The position of Figure 2 changed and modified.

Comment-10: Line 270 Aurelin peptide isolated from Aurelia aurita showed antimicrobial activity against Gram-positive bacteria by blocking the potassium channel [49] (Table 1).  Table 1 should be changed to Table 2.

 [Response] The position of the Tables has been changed and modified.

Comment-11: Line 286 Myticin C, isolated from Mytilus galloprovincialis is an antiviral peptide that interferes with viral replication (Table 1). Myticin C and above compounds cannot find in Table 1. Please check.

 [Response] I'd like to thank you for informing us about the inadvertent error in the manuscript. The table numbering has been corrected. Myticin C is included in Table 2.   

Comment-12: Figure 4 should be explained and listed in appropriate section.

 [Response] The correction has been included and Figure 4 has been explained and listed in the appropriate section.

Comment-13: Figure 4. Current status of the natural AMPs to date as per AMP database (https://aps.unmc.edu/).

 [Response] We addressed the reviewers' concerns.

Comment-14: Line 377 8.3. Hinders production of extracellular matrix. You may consider hindering cell propagation through extracellular matrix interference 

 [Response] I'd like to thank you for informing us about the inadvertent error in the manuscript. We have rewritten the sentences (line 491).